# STRATEGIES FOR PRE-TRAINING GRAPH NEURAL NETWORKS

**Weihua Hu**[1]*, **Bowen Liu**[2]*, **Joseph Gomes**[4], **Marinka Zitnik**[5],
**Percy Liang**[1], **Vijay Pande**[3], **Jure Leskovec**[1]
[1]Department of Computer Science, [2]Chemistry, [3]Bioengineering, Stanford University,
[4]Department of Chemical and Biochemical Engineering, The University of Iowa,
[5]Department of Biomedical Informatics, Harvard University
`{weihuahu,liubowen,pliang,jure}@cs.stanford.edu`,
`joe-gomes@uiowa.edu, marinka@hms.harvard.edu, pande@stanford.edu`

## ABSTRACT

Many applications of machine learning require a model to make accurate predictions on test examples that are distributionally different from training ones, while task-specific labels are scarce during training. An effective approach to this challenge is to pre-train a model on related tasks where data is abundant, and then fine-tune it on a downstream task of interest. While pre-training has been effective in many language and vision domains, it remains an open question how to effectively use pre-training on graph datasets. In this paper, we develop a new strategy and self-supervised methods for pre-training Graph Neural Networks (GNNs). The key to the success of our strategy is to pre-train an expressive GNN at the level of individual nodes as well as entire graphs so that the GNN can learn useful local and global representations simultaneously. We systematically study pre-training on multiple graph classification datasets. We find that naïve strategies, which pre-train GNNs at the level of either entire graphs or individual nodes, give limited improvement and can even lead to negative transfer on many downstream tasks. In contrast, our strategy avoids negative transfer and improves generalization significantly across downstream tasks, leading up to 9.4% absolute improvements in ROC-AUC over non-pre-trained models and achieving state-of-the-art performance for molecular property prediction and protein function prediction.

## 1 INTRODUCTION

Transfer learning refers to the setting where a model, initially trained on some tasks, is re-purposed on different but related tasks. Deep transfer learning has been immensely successful in computer vision (Donahue et al., 2014; Girshick et al., 2014; Zeiler & Fergus, 2014) and natural language processing (Devlin et al., 2019; Peters et al., 2018; Mikolov et al., 2013). Despite being an effective approach to transfer learning, few studies have generalized pre-training to graph data.

Pre-training has the potential to provide an attractive solution to the following two fundamental challenges with learning on graph datasets (Pan & Yang, 2009; Hendrycks et al., 2019): First, task-specific labeled data can be extremely scarce. This problem is exacerbated in important graph datasets from scientific domains, such as chemistry and biology, where data labeling (*e.g.*, biological experiments in a wet laboratory) is resource- and time-intensive (Zitnik et al., 2018). Second, graph data from real-world applications often contain out-of-distribution samples, meaning that graphs in the training set are structurally very different from graphs in the test set. Out-of-distribution prediction is common in real-world graph datasets, for example, when one wants to predict chemical properties of a brand-new, just synthesized molecule, which is different from all molecules synthesized so far, and thereby different from all molecules in the training set.

However, pre-training on graph datasets remains a hard challenge. Several key studies (Xu et al., 2017; Ching et al., 2018; Wang et al., 2019) have shown that successful transfer learning is not only a

---

*Equal contribution. Project website, data and code: `http://snap.stanford.edu/gnn-pretrain`

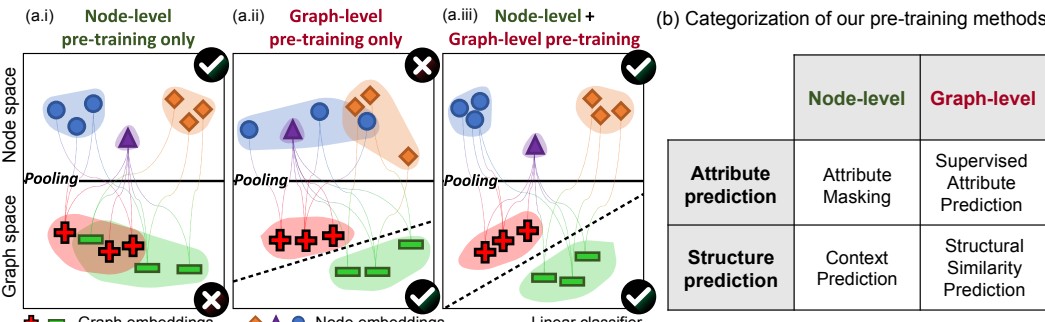

Figure 1: **(a.i)** When only node-level pre-training is used, nodes of different shapes (semantically different nodes) can be well separated, however, node embeddings are not composable, and thus resulting graph embeddings (denoted by their classes, + and −) that are created by pooling node-level embeddings are not separable. **(a.ii)** With graph-level pre-training only, graph embeddings are well separated, however the embeddings of individual nodes do not necessarily capture their domain-specific semantics. **(a.iii)** High-quality node embeddings are such that nodes of different types are well separated, while at the same time, the embedding space is also composable. This allows for accurate and robust representations of entire graphs and enables robust transfer of pre-trained models to a variety of downstream tasks. **(b)** Categorization of pre-training methods for GNNs. Crucially, our methods, *i.e.*, Context Prediction, Attribute Masking, and graph-level supervised pre-training (Supervised Attribute Prediction) enable both node-level and graph-level pre-training.

matter of increasing the number of labeled pre-training datasets that are from the same domain as the downstream task. Instead, it requires substantial domain expertise to carefully select examples and target labels that are correlated with the downstream task of interest. Otherwise, the transfer of knowledge from related pre-training tasks to a new downstream task can harm generalization, which is known as *negative transfer* (Rosenstein et al., 2005) and significantly limits the applicability and reliability of pre-trained models.

**Present work.** Here, we focus on pre-training as an approach to transfer learning in Graph Neural Networks (GNNs) (Kipf & Welling, 2017; Hamilton et al., 2017a; Ying et al., 2018b; Xu et al., 2019; 2018) for graph-level property prediction. Our work presents two key contributions. (1) We conduct the first systematic large-scale investigation of strategies for pre-training GNNs. For that, we build two large new pre-training datasets, which we share with the community: a chemistry dataset with 2 million graphs and a biology dataset with 395K graphs. We also show that large domain-specific datasets are crucial to investigate pre-training and that existing downstream benchmark datasets are too small to evaluate models in a statistically reliable way. (2) We develop an effective pre-training strategy for GNNs and demonstrate its effectiveness and its ability for out-of-distribution generalization on hard transfer-learning problems.

In our systematic study, we show that pre-training GNNs does not always help. Naïve pre-training strategies can lead to negative transfer on many downstream tasks. Strikingly, a seemingly strong pre-training strategy (*i.e.*, graph-level multi-task supervised pre-training using a state-of-the-art graph neural network architecture for graph-level prediction tasks) only gives marginal performance gains. Furthermore, this strategy even leads to negative transfer on many downstream tasks (2 out of 8 molecular datasets and 13 out of 40 protein prediction tasks).

We develop an effective strategy for pre-training GNNs. The key idea is to use easily accessible node-level information and encourage GNNs to capture domain-specific knowledge about nodes and edges, in addition to graph-level knowledge. This helps the GNN to learn useful representations at both global and local levels (Figure 1 (a.iii)), and is crucial to be able to generate graph-level representations (which are obtained by pooling node representations) that are robust and transferable to diverse downstream tasks (Figure 1). Our strategy is in contrast to naïve strategies that either leverage only at graph-level properties (Figure 1 (a.ii)) or node-level properties (Figure 1 (a.i)).

Empirically, our pre-training strategy used together with the most expressive GNN architecture, GIN (Xu et al., 2019), yields state-of-the-art results on benchmark datasets and avoids negative transfer across downstream tasks we tested. It significantly improves generalization performance

across downstream tasks, yielding up to 9.4% higher average ROC-AUC than non-pre-trained GNNs, and up to 5.2% higher average ROC-AUC compared to GNNs with the extensive graph-level multi-task supervised pre-training. Furthermore, we find that the most expressive architecture, GIN, benefits more from pre-training compared to those with less expressive power (*e.g.*, GCN (Kipf & Welling, 2017), GraphSAGE (Hamilton et al., 2017b) and GAT (Velickovic et al., 2018)), and that pre-training GNNs leads to orders-of-magnitude faster training and convergence in the fine-tuning stage.

## 2 Preliminaries of Graph Neural Networks

We first formalize supervised learning of graphs and provide an overview of GNNs (Gilmer et al., 2017). Then, we briefly review methods for unsupervised graph representation learning.

**Supervised learning of graphs.** Let $G = (V, E)$ denote a graph with node attributes $X_v$ for $v \in V$ and edge attributes $e_{uv}$ for $(u, v) \in E$. Given a set of graphs $\{G_1, \ldots, G_N\}$ and their labels $\{y_1, \ldots, y_N\}$, the task of graph supervised learning is to learn a representation vector $h_G$ that helps predict the label of an entire graph $G$, $y_G = g(h_G)$. For example, in molecular property prediction, $G$ is a molecular graph, where nodes represent atoms and edges represent chemical bonds, and the label to be predicted can be toxicity or enzyme binding.

**Graph Neural Networks (GNNs).** GNNs use the graph connectivity as well as node and edge features to learn a representation vector (*i.e.*, embedding) $h_v$ for every node $v \in G$ and a vector $h_G$ for the entire graph $G$. Modern GNNs use a neighborhood aggregation approach, where representation of node $v$ is iteratively updated by aggregating representations of $v$'s neighboring nodes and edges (Gilmer et al., 2017). After $k$ iterations of aggregation, $v$'s representation captures the structural information within its $k$-hop network neighborhood. Formally, the $k$-th layer of a GNN is:

$$h_v^{(k)} = \text{COMBINE}^{(k)}\left(h_v^{(k-1)}, \text{AGGREGATE}^{(k)}\left(\left\{\left(h_v^{(k-1)}, h_u^{(k-1)}, e_{uv}\right): u \in \mathcal{N}(v)\right\}\right)\right), \quad (2.1)$$

where $h_v^{(k)}$ is the representation of node $v$ at the $k$-th iteration/layer, $e_{uv}$ is the feature vector of edge between $u$ and $v$, and $\mathcal{N}(v)$ is a set neighbors of $v$. We initialize $h_v^{(0)} = X_v$.

**Graph representation learning.** To obtain the entire graph's representation $h_G$, the READOUT function pools node features from the final iteration $K$,

$$h_G = \text{READOUT}\left(\left\{h_v^{(K)} \mid v \in G\right\}\right). \quad (2.2)$$

READOUT is a permutation-invariant function, such as averaging or a more sophisticated graph-level pooling function (Ying et al., 2018b; Zhang et al., 2018).

## 3 Strategies for pre-training Graph Neural Networks

At the technical core of our pre-training strategy is the notion to pre-train a GNN *both* at the level of individual nodes as well as entire graphs. This notion encourages the GNN to capture domain-specific semantics at both levels, as illustrated in Figure 1 (a.iii). This is in contrast to straightforward but limited pre-training strategies that either only use pre-training to predict properties of entire graphs (Figure 1 (a.ii)) or only use pre-training to predict properties of individual nodes (Figure 1 (a.i)).

In the following, we first describe our node-level pre-training approach (Section 3.1) and then graph-level pre-training approach (Section 3.2). Finally, we describe the full pre-training strategy in Section 3.3.

### 3.1 Node-level pre-training

For node-level pre-training of GNNs, our approach is to use easily-accessible unlabeled data to capture domain-specific knowledge/regularities in the graph. Here we propose two self-supervised methods, Context Prediction and Attribute Masking.

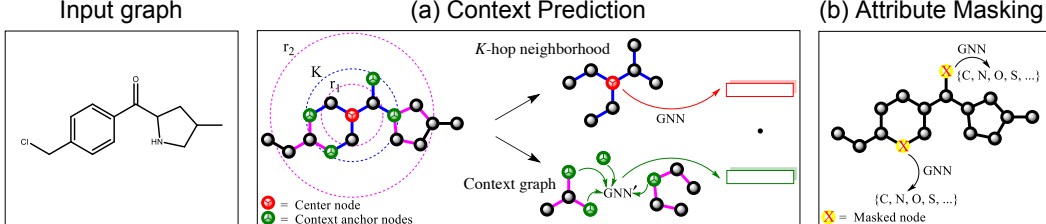

Figure 2: Illustration of our node-level methods, Context Prediction and Attribute Masking for pre-training GNNs. **(a)** In Context Prediction, the subgraph is a $K$-hop neighborhood around a selected center node, where $K$ is the number of GNN layers and is set to 2 in the figure. The context is defined as the surrounding graph structure that is between $r_1$- and $r_2$-hop from the center node, where we use $r_1 = 1$ and $r_2 = 4$ in the figure. **(b)** In Attribute Masking, the input node/edge attributes (*e.g.*, atom type in the molecular graph) are randomly masked, and the GNN is asked to predict them.

### 3.1.1 CONTEXT PREDICTION: EXPLOITING DISTRIBUTION OF GRAPH STRUCTURE

In Context Prediction, we use subgraphs to predict their surrounding graph structures. Our goal is to pre-train a GNN so that it maps nodes appearing in similar structural contexts to nearby embeddings (Rubenstein & Goodenough, 1965; Mikolov et al., 2013).

**Neighborhood and context graphs.** For every node $v$, we define $v$'s neighborhood and context graphs as follows. $K$-*hop neighborhood* of $v$ contains all nodes and edges that are at most $K$-hops away from $v$ in the graph. This is motivated by the fact that a $K$-layer GNN aggregates information across the $K$-th order neighborhood of $v$, and thus node embedding $h_v^{(K)}$ depends on nodes that are at most $K$-hops away from $v$. We define *context graph* of node $v$ as graph structure that surrounds $v$'s neighborhood. The context graph is described by two hyperparameters, $r_1$ and $r_2$, and it represents a subgraph that is between $r_1$-hops and $r_2$-hops away from $v$ (*i.e.*, it is a ring of width $r_2 - r_1$). Examples of neighborhood and context graphs are shown in Figure 2 (a). We require $r_1 < K$ so that some nodes are shared between the neighborhood and the context graph, and we refer to those nodes as *context anchor nodes*. These anchor nodes provide information about how the neighborhood and context graphs are connected with each other.

**Encoding context into a fixed vector using an auxiliary GNN.** Directly predicting the context graph is intractable due to the combinatorial nature of graphs. This is different from natural language processing, where words come from a fixed and finite vocabulary. To enable context prediction, we encode context graphs as *fixed-length vectors*. To this end, we use an auxiliary GNN, which we refer to as the *context GNN*. As depicted in Figure 2 (a), we first apply the context GNN (denoted as GNN$'$ in Figure 2 (a)) to obtain node embeddings in the context graph. We then average embeddings of *context anchor nodes* to obtain a fixed-length context embedding. For node $v$ in graph $G$, we denote its corresponding context embedding as $c_v^G$.

**Learning via negative sampling.** We then use negative sampling (Mikolov et al., 2013; Ying et al., 2018a) to jointly learn the main GNN and the context GNN. The main GNN encodes neighborhoods to obtain node embeddings. The context GNN encodes context graphs to obtain context embeddings. In particular, the learning objective of Context Prediction is a binary classification of whether a particular neighborhood and a particular context graph belong to the same node:

$$\sigma\left(h_v^{(K)\top} c_{v'}^{G'}\right) \approx \mathbf{1}\{v \text{ and } v' \text{ are the same nodes}\}, \tag{3.1}$$

where $\sigma(\cdot)$ is the sigmoid function, and $\mathbf{1}(\cdot)$ is the indicator function. We either let $v' = v$ and $G' = G$ (*i.e.*, a positive neighborhood-context pair), or we randomly sample $v'$ from a randomly chosen graph $G'$ (*i.e.*, a negative neighborhood-context pair). We use a negative sampling ratio of 1 (one negative pair per one positive pair), and use the negative log likelihood as the loss function. After pre-training, the main GNN is retained as our pre-trained model

### 3.1.2 ATTRIBUTE MASKING: EXPLOITING DISTRIBUTION OF GRAPH ATTRIBUTES

In Attribute Masking, we aim to capture domain knowledge by learning the regularities of the node/edge attributes distributed over graph structure.

**Masking node and edges attributes.** Attribute Masking pre-training works as follows: We mask node/edge attributes and then we let GNNs predict those attributes (Devlin et al., 2019) based on neighboring structure. Figure 2 (b) illustrates our proposed method when applied to a molecular graph. Specifically, We randomly mask input node/edge attributes, for example atom types in molecular graphs, by replacing them with special masked indicators. We then apply GNNs to obtain the corresponding node/edge embeddings (edge embeddings can be obtained as a sum of node embeddings of the edge's end nodes). Finally, a linear model is applied on top of embeddings to predict a masked node/edge attribute. Different from Devlin et al. (2019) that operates on sentences and applies message passing over the fully-connected graph of tokens, we operate on non-fully-connected graphs and aim to capture the regularities of node/edge attributes distributed over different graph structures. Furthermore, we allow masking edge attributes, going beyond masking node attributes.

Our node and edge attribute masking method is especially beneficial for richly-annotated graphs from scientific domains. For example, (1) in molecular graphs, the node attributes correspond to atom types, and capturing how they are distributed over the graphs enables GNNs to learn simple chemistry rules such as valency, as well as potentially more complex chemistry phenomenon such as the electronic or steric properties of functional groups. Similarly, (2) in protein-protein interaction (PPI) graphs, the edge attributes correspond to different kinds of interactions between a pair of proteins. Capturing how these attributes distribute across the PPI graphs enables GNNs to learn how different interactions relate and correlate with each other.

## 3.2 GRAPH-LEVEL PRE-TRAINING

We aim to pre-train GNNs to generate useful graph embeddings composed of the meaningful node embeddings obtained by methods in Section 3.1. Our goal is to ensure both node and graph embeddings are of high-quality so that graph embeddings are robust and transferable across downstream tasks, as illustrated in Figure 1 (a.iii). Additionally, there are two options for graph-level pre-training, as shown in Figure 1 (b): making predictions about domain-specific attributes of entire graphs (*e.g.*, supervised labels), or making predictions about graph structure.

### 3.2.1 SUPERVISED GRAPH-LEVEL PROPERTY PREDICTION

As the graph-level representation $h_G$ is directly used for fine-tuning on downstream prediction tasks, it is desirable to directly encode domain-specific information into $h_G$.

We inject graph-level domain-specific knowledge into our pretrained embeddings by defining supervised graph-level prediction tasks. In particular, we consider a practical method to pre-train graph representations: graph-level multi-task supervised pre-training to jointly predict a diverse set of supervised labels of individual graphs. For example, in molecular property prediction, we can pre-train GNNs to predict essentially all the properties of molecules that have been experimentally measured so far. In protein function prediction, where the goal is predict whether a given protein has a given functionality, we can pre-train GNNs to predict the existence of diverse protein functions that have been validated so far. In our experiments in Section 5, we prepare a diverse set of supervised tasks (up to 5000 tasks) to simulate these practical scenarios. Further details of the supervised tasks and datasets are described in Section 5.1. To jointly predict many graph properties, where each property corresponds to a binary classification task, we apply linear classifiers on top of graph representations.

Importantly, naïvely performing the extensive multi-task graph-level pre-training alone can fail to give transferable graph-level representations, as empirically demonstrated in Section 5. This is because some supervised pre-training tasks might be unrelated to the downstream task of interest and can even hurt the downstream performance (negative transfer). One solution would be to select "truly-relevant" supervised pre-training tasks and pre-train GNNs only on those tasks. However, such a solution is extremely costly since selecting the relevant tasks requires significant domain expertise and pre-training needs to be performed separately for different downstream tasks.

To alleviate this issue, our key insight is that the multi-task supervised pre-training only provides graph-level supervision; thus, local node embeddings from which the graph-level embeddings are created may not be meaningful, as illustrated in Figure 1 (a.ii). Such non-useful node embeddings can exacerbate the problem of negative transfer because many different pre-training tasks can more easily interfere with each other in the node embedding space. Motivated by this, our pre-training strategy is to first regularize GNNs at the level of individual nodes via node-level pre-training methods described in Section 3.1, before performing graph-level pre-training. As we demonstrate empirically, the combined strategy produces much more transferable graph representations and robustly improves downstream performance without expert selection of supervised pre-training tasks.

### 3.2.2 STRUCTURAL SIMILARITY PREDICTION

A second approach is to define a graph-level predictive task where the goal would be to model the structural similarity of two graphs. Examples of such tasks include modeling the graph edit distance (Bai et al., 2019) or predicting graph structure similarity (Navarin et al., 2018). However, finding the ground truth graph distance values is a difficult problem, and in large datasets there is a quadratic number of graph pairs to consider. Therefore, while this type of pre-training is also very natural, it is beyond the scope of this paper and we leave its investigation for future work.

### 3.3 OVERVIEW: PRE-TRAINING GNNs AND FINE-TUNING FOR DOWNSTREAM TASKS

Altogether, our pre-training strategy is to first perform node-level self-supervised pre-training (Section 3.1) and then graph-level multi-task supervised pre-training (Section 3.2). When the GNN pre-training is finished, we fine-tune the pre-trained GNN model on downstream tasks. Specifically, we add linear classifiers on top of graph-level representations to predict downstream graph labels. The full model, *i.e.*, the pre-trained GNN and downstream linear classifiers, is subsequently fine-tuned in an end-to-end manner. Time-complexity analysis is provided in Appendix F, where we show that our pre-training methods incur little computational overhead to forward computation in GNNs.

## 4 FURTHER RELATED WORK

There is rich literature on unsupervised representation learning of individual *nodes* within graphs, which broadly falls into two categories. In the first category are methods that use local random walk-based objectives (Grover & Leskovec, 2016; Perozzi et al., 2014; Tang et al., 2015) and methods that reconstruct a graph's adjacency matrix, *e.g.*, by predicting edge existence (Hamilton et al., 2017a; Kipf & Welling, 2016). In the second category are methods, such as Deep Graph Infomax (Veličković et al., 2019), that train a node encoder that maximizes mutual information between local node representations and a pooled global graph representation. All these methods encourage nearby nodes to have similar embeddings and were originally proposed and evaluated for node classification and link prediction. This, however, can be sub-optimal for graph-level prediction tasks, where capturing *structural* similarity of local neighborhoods is often more important than capturing the positional information of nodes within a graph (You et al., 2019; Rogers & Hahn, 2010; Yang et al., 2014). Our approach thus considers both the node-level as well as graph-level pretraining tasks and as we show in our experiments, it is essential to use both types of tasks in order for pretrained models to achieve good performance.

A number of recent works have also explored how node embeddings generalize across tasks (Jaeger et al., 2018; Zhou et al., 2018; Chakravarti, 2018; Narayanan et al., 2016). However, all of these methods use distinct node embeddings for different substructures and do not share any parameters. Thus, they are inherently transductive, cannot transfer between datasets, cannot be fine-tuned in an end-to-end manner, and cannot capture large and diverse neighborhoods/contexts due to data sparsity. Our approach addresses all these challenges by developing pre-training methods for GNNs that use shared parameters to encode the the graph-level as well as node-level dependencies and structures.

## 5 EXPERIMENTS

### 5.1 DATASETS

We consider two domains; molecular property prediction in chemistry and protein function prediction in biology. We release the new datasets at: `http://snap.stanford.edu/gnn-pretrain`.

**Pre-training datasets.** For the chemistry domain, we use 2 million unlabeled molecules sampled from the ZINC15 database (Sterling & Irwin, 2015) for node-level self-supervised pre-training. For graph-level multi-task supervised pre-training, we use a preprocessed ChEMBL dataset (Mayr et al., 2018; Gaulton et al., 2011), containing 456K molecules with 1310 kinds of diverse and extensive biochemical assays. For the biology domain, we use 395K unlabeled protein ego-networks derived from PPI networks of 50 species (*e.g.*, humans, yeast, zebra fish) for node-level self-supervised pre-training. For graph-level multi-task supervised pre-training, we use 88K labeled protein ego-networks to jointly predict 5000 *coarse-grained* biological functions (*e.g.*, cell apoptosis, cell proliferation).

**Downstream classification datasets.** For the chemistry domain, we considered classical graph classification benchmarks (MUTAG, PTC molecule datasets) (Kersting et al., 2016; Xu et al., 2019) as our downstream tasks, but found that they are too small (188 and 344 examples for MUTAG and PTC) to evaluate different methods in a statistically meaningful way (see Appendix B for the results and discussion). Because of this, as our downstream tasks, we decided to use 8 larger binary classification datasets contained in MoleculeNet (Wu et al., 2018), a recently-curated benchmark for molecular property prediction. The dataset statistics are summarized in Table 1. For the biology domain, we compose our PPI networks from Zitnik et al. (2019), consisting of 88K proteins from 8 different species, where the subgraphs centered at a protein of interest (*i.e.*, ego-networks) are used to predict their biological functions. Our downstream task is to predict 40 *fine-grained* biological functions[1] that correspond to 40 binary classification tasks. In contrast to existing PPI datasets (Hamilton et al., 2017a), our dataset is larger and spans multiple species (*i.e.*, not only humans), which makes it a suitable benchmark for evaluating out-of-distribution prediction. Additional details about datasets and features of molecule/PPI graphs are in Appendices C and D.

**Dataset splitting.** In many applications, conventional random split is overly optimistic and does not simulate the real-world use case, where test graphs can be structurally different from training graphs (Wu et al., 2018; Zitnik et al., 2019). To reflect the actual use case, we split the downstream data in the following ways to evaluate the models' *out-of-distribution generalization*. In the chemistry domain, we use *scaffold split* (Ramsundar et al., 2019), where we split molecules according to their scaffold (molecular substructure). In the biology domain, we use *species split*, where we predict functions of proteins from new species. Details are in Appendix E. Furthermore, to prevent data leakage, all test graphs used for performance evaluation are removed from the graph-level supervised pre-training datasets.

### 5.2 EXPERIMENTAL SETUP

We thoroughly compare our pre-training strategy with two naïve baseline strategies: (i) extensive supervised multi-task pre-training on relevant graph-level tasks, and (ii) node-level self-supervised pre-training.

**GNN architectures.** We mainly study Graph Isomorphism Networks (GINs) (Xu et al., 2019), the most expressive and state-of-the-art GNN architecture for graph-level prediction tasks. We also experimented with other popular architectures that are less expressive: GCN (Kipf & Welling, 2016), GAT (Veličković et al., 2019), and GraphSAGE (with mean neighborhood aggregation) (Hamilton et al., 2017b). We select the following hyper-parameters that performed well across all downstream tasks in the validation sets: 300 dimensional hidden units, 5 GNN layers ($K = 5$), and average pooling for the READOUT function. Additional details can be found in Appendix A.

**Pre-training.** For Context Prediction illustrated in Figure 2 (a), on molecular graphs, we define context graphs by setting inner radius $r_1 = 4$. On PPI networks whose diameters are often smaller than 5, we use $r_1 = 1$, which works well empirically despite the large overlap between the neighborhood and context subgraphs. For both molecular and PPI graphs, we let outer radius $r_2 = r_1 + 3$, and

---

[1]Fine-grained labels are harder to obtain than coarse-grained labels; the latter are used for pre-training.

| Dataset | | BBBP | Tox21 | ToxCast | SIDER | ClinTox | MUV | HIV | BACE | Average |
|---|---|---|---|---|---|---|---|---|---|---|
| # Molecules | | 2039 | 7831 | 8575 | 1427 | 1478 | 93087 | 41127 | 1513 | / |
| # Binary prediction tasks | | 1 | 12 | 617 | 27 | 2 | 17 | 1 | 1 | / |
| Pre-training strategy | | Out-of-distribution prediction (scaffold split) | | | | | | | | |
| Graph-level | Node-level | | | | | | | | | |
| – | – | 65.8 ±4.5 | 74.0 ±0.8 | 63.4 ±0.6 | 57.3 ±1.6 | 58.0 ±4.4 | 71.8 ±2.5 | 75.3 ±1.9 | 70.1 ±5.4 | 67.0 |
| – | Infomax | **68.8 ±0.8** | 75.3 ±0.5 | 62.7 ±0.4 | 58.4 ±0.8 | 69.9 ±3.0 | 75.3 ±2.5 | 76.0 ±0.7 | 75.9 ±1.6 | 70.3 |
| – | EdgePred | 67.3 ±2.4 | 76.0 ±0.6 | 64.1 ±0.6 | 60.4 ±0.7 | 64.1 ±3.7 | 74.1 ±2.1 | 76.3 ±1.0 | 79.9 ±0.9 | 70.3 |
| – | AttrMasking | 64.3 ±2.8 | 76.7 ±0.4 | 64.2 ±0.5 | 61.0 ±0.7 | 71.8 ±4.1 | 74.7 ±1.4 | 77.2 ±1.1 | 79.3 ±1.6 | 71.1 |
| – | ContextPred | 68.0 ±2.0 | 75.7 ±0.7 | 63.9 ±0.6 | 60.9 ±0.6 | 65.9 ±3.8 | 75.8 ±1.7 | 77.3 ±1.0 | 79.6 ±1.2 | 70.9 |
| Supervised | – | 68.3 ±0.7 | 77.0 ±0.3 | 64.4 ±0.4 | 62.1 ±0.5 | 57.2 ±2.5 | 79.4 ±1.3 | 74.4 ±1.2 | 76.9 ±1.0 | 70.0 |
| Supervised | Infomax | 68.0 ±1.8 | 77.8 ±0.3 | 64.9 ±0.7 | 60.9 ±0.6 | **71.2 ±2.8** | 81.3 ±1.4 | 77.8 ±0.9 | 80.1 ±0.9 | 72.8 |
| Supervised | EdgePred | 66.6 ±2.2 | **78.3 ±0.3** | **66.5 ±0.3** | 63.3 ±0.9 | 70.9 ±4.6 | 78.5 ±2.4 | 77.5 ±0.8 | 79.1 ±3.7 | 72.6 |
| Supervised | AttrMasking | 66.5 ±2.5 | 77.9 ±0.4 | 65.1 ±0.3 | **63.9 ±0.9** | **73.7 ±2.8** | **81.2 ±1.9** | 77.1 ±1.2 | 80.3 ±0.9 | 73.2 |
| Supervised | ContextPred | **68.7 ±1.3** | 78.1 ±0.6 | 65.7 ±0.6 | 62.7 ±0.8 | **72.6 ±1.5** | **81.3 ±2.1** | 79.9 ±0.7 | **84.5 ±0.7** | **74.2** |

Table 1: **Test ROC-AUC (%) performance on molecular prediction benchmarks using different pre-training strategies with GIN.** The rightmost column averages the mean of test performance across the 8 datasets. The best result for each dataset and comparable results (*i.e.*, results within one standard deviation from the best result) are bolded. The shaded cells indicate negative transfer, *i.e.*, ROC-AUC of a pre-trained model is worse than that of a non-pre-trained model. Notice that node- as well as graph-level pretraining are essential for good performance.

| | Chemistry | | | Biology | | |
|---|---|---|---|---|---|---|
| | Non-pre-trained | Pre-trained | Gain | Non-pre-trained | Pre-trained | Gain |
| GIN | 67.0 | **74.2** | **+7.2** | 64.8 ± 1.0 | **74.2 ± 1.5** | **+9.4** |
| GCN | **68.9** | 72.2 | +3.4 | 63.2 ± 1.0 | 70.9 ± 1.7 | +7.7 |
| GraphSAGE | 68.3 | 70.3 | +2.0 | 65.7 ± 1.2 | 68.5 ± 1.5 | +2.8 |
| GAT | 66.8 | 60.3 | -6.5 | **68.2 ± 1.1** | 67.8 ± 3.6 | -0.4 |

Table 2: **Test ROC-AUC (%) performance of different GNN architectures with and without pre-training.** Without pre-training, the less expressive GNNs give slightly better performance than the most expressive GIN because of their smaller model complexity in a low data regime. However, with pre-training, the most expressive GIN is properly regularized and dominates the other architectures. For results split by chemistry datasets, see Table 4 in Appendix H. Pre-training strategy for chemistry data: Context Prediction + Graph-level supervised pre-training; pre-training strategy for biology data: Attribute Masking + Graph-level supervised pre-training.

use a 3-layer GNN to encode the context structure. For Attribute Masking shown in Figure 2 (b), we randomly mask 15% of node (for molecular graphs) or edge attributes (for PPI networks) for prediction. As baselines for node-level self-supervised pre-training, we adopt the original Edge Prediction (denoted by EdgePred) (Hamilton et al., 2017a) and Deep Graph Infomax (denoted by Infomax) (Veličković et al., 2019) implementations. Further details are provided in Appendix G.

## 5.3 RESULTS

We report results for molecular property prediction and protein function prediction in Tables 2 and 1 and Figure 3. Our systematic study suggests the following trends:

**Observation (1):** Table 2 shows that the most expressive GNN architecture (GIN), when pre-trained, achieves the best performance across domains and datasets. Compared with gains of pre-training achieved by GIN architecture, gains of pre-training using less expressive GNNs (GCN, GraphSAGE, and GAT) are smaller and can sometimes even be negative (Table 2). This finding confirms previous observations (*e.g.*, Erhan et al. (2010)) that using an expressive model is crucial to fully utilize pre-training, and that pre-training can even hurt performance when used on models with limited expressive power, such as GCN, GraphSAGE, and GAT.

**Observation (2):** As seen from the shaded cells of Table 1 and highlighted region in the middle panel of Figure 3, the strong baseline strategy that performs extensive graph-level multi-task supervised pre-training of GNNs gives surprisingly limited performance gain and yields negative transfer on many downstream tasks (2 out of 8 datasets in molecular prediction, and 13 out of 40 tasks in protein function prediction).

**Observation (3):** From the upper half of Table 1 and the left panel of Figure 3, we see that another baseline strategy, which only performs node-level self-supervised pre-training, also gives limited performance improvement and is comparable to the graph-level multi-task supervised pre-training baseline.

**Observation (4):** From the lower half of Table 1 and the right panel of Figure 3, we see that our pre-training strategy of combining graph-level multi-task supervised and node-level self-supervised pre-training avoids negative transfer across downstream datasets and achieves best performance.

**Observation (5):** Furthermore, from Table 1 and the left panel of Figure 3, we see that our strategy gives significantly better predictive performance than the two baseline pre-training strategies as well as non-pre-trained models, achieving state-of-the-art performance.

Specifically, in the **chemistry** datasets, we see from Table 1 that our Context Prediction + Graph-level multi-task supervised pre-training strategy gives the most promising performance, leading to an increase in average ROC-AUC of 7.2% over non-pre-trained baseline and 4.2% over graph-level multi-task supervised pre-trained baseline. On the HIV dataset, where a number of recent works (Wu et al., 2018; Li et al., 2017; Ishiguro et al., 2019) have reported performance on the same scaffold split and using the same protocol, our best pre-trained model (ContextPred + Supervised) achieves state-of-the-art performance. In particular, we achieved a ROC-AUC score of 79.9%, while best-performing graph models in Wu et al. (2018), Li et al. (2017), and Ishiguro et al. (2019) had ROC-AUC scores of 76.3%, 77.6%, and 76.2%, respectively.

Also, in the **biology** datasets, which we have built in this work, we see from the left panel of Figure 3 that our Attribute Masking + Graph-level multi-task supervised pre-training strategy achieves the best predictive performance compared to other baseline strategies *across almost all* 40 downstream prediction tasks (the right panel of Figure 3). On average, our strategy improves ROC-AUC by 9.4% over non-pre-trained baseline and 5.2% over graph-level multi-task supervised pre-trained baseline, again achieving state-of-the-art performance.

**Observation (6):** In the chemistry domain, we also report performance on classic benchmarks (MUTAG, PTC molecule datasets) in Appendix B. However, as mentioned in Section 5.1, the extremely small dataset sizes make these benchmarks unsuitable to compare different methods in a statistically reliable way.

**Observation (7):** Beyond predictive performance improvement, Figure 4 shows that our pre-trained models achieve orders-of-magnitude faster training and validation convergence than non-pre-trained models. For example, on the MUV dataset, it took 1 hour for the non-pre-trained GNN to get 74.9% validation ROC-AUC, while it took only 5 minutes for our pre-trained GNN to get 85.3% validation ROC-AUC. The same trend holds across the downstream datasets we used, as shown in Figure 5 in Appendix I. We emphasize that pre-training is a one-time-effort. Once the model is pre-trained, it can be used for any number of downstream tasks to improve performance with little training time.

As a final remark, in our preliminary experiments, we performed the Attribute Masking and Context Prediction simultaneously to pre-train GNNs. That approach did not improve performance in our experiments. We leave a thorough analysis of the approach for future work.

## 6 CONCLUSIONS AND FUTURE WORK

We developed a novel strategy for pre-training GNNs. Crucial to the success of our strategy is to consider both node-level and graph-level pre-training in combination with an expressive GNN. This ensures that node embeddings capture local neighborhood semantics that are pooled together to obtain meaningful graph-level representations, which, in turn, are used for downstream tasks. Experiments on multiple datasets, diverse downstream tasks and different GNN architectures show that the new pre-training strategy achieves consistently better out-of-distribution generalization than non-pre-trained models.

Our work makes an important step toward transfer learning on graphs and addresses the issue of negative transfer observed in prior studies. There are many interesting avenues for future work. For example, further increasing generalization by improving GNN architectures as well as pre-training and fine-tuning approaches, is a fruitful direction. Investigating what pre-trained models have learned would also be useful to aid scientific discovery (Tshitoyan et al., 2019). Finally, it would be interesting

| Pre-training strategy | | Out-of-dist. |
| --- | --- | --- |
| Graph-level | Node-level | (species split) |
| – | – | 64.8 ±1.0 |
| – | Infomax | 64.1 ±1.5 |
| – | EdgePred | 65.7 ±1.3 |
| – | ContextPred | 65.2 ±1.6 |
| – | AttrMasking | 64.4 ±1.3 |
| Supervised | – | 69.0 ±2.4 |
| Supervised | Infomax | 72.8 ±1.5 |
| Supervised | EdgePred | 72.3 ±1.4 |
| Supervised | ContextPred | 73.8 ±1.0 |
| Supervised | AttrMasking | **74.2 ±1.5** |

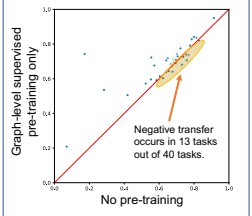
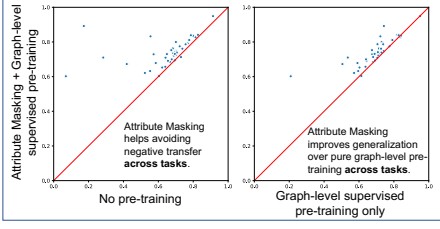

Figure 3: **Test ROC-AUC of protein function prediction using different pre-training strategies with GIN. (Left)** Test ROC-AUC scores (%) obtained by different pre-training strategies, where the scores are averaged over the 40 fine-grained prediction tasks. **(Middle and right):** Scatter plot comparisons of ROC-AUC scores for a pair of pre-training strategies on the 40 *individual* downstream tasks. Each point represents a particular individual downstream task. **(Middle):** There are many *individual* downstream tasks where graph-level multi-task supervised pre-trained model performs worse than non-pre-trained model, indicating negative transfer. **(Right):** When the graph-level multi-task supervised pre-training and Attribute Masking are combined, negative transfer is avoided across downstream tasks. The performance also improves over pure graph-level supervised pre-training.

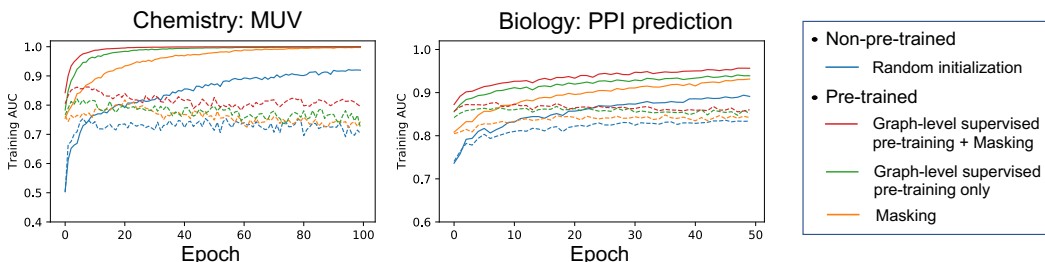

Figure 4: **Training and validation curves of different pre-training strategies on GINs.** Solid and dashed lines indicate training and validation curves, respectively.

to apply our methods to other domains, *e.g.*, physics, material science, and structural biology, where many problems are defined over graphs representing interactions of *e.g.*, atoms, particles, and amino acids.

## ACKNOWLEDGMENTS

We thank Camilo Ruiz, Rex Ying, Zhenqin Wu, Shantao Li, Srijan Kumar, Hongwei Wang, and Robin Jia for their helpful discussion. W.H is supported by Funai Overseas Scholarship and Masason Foundation Fellowship. J.L is a Chan Zuckerberg Biohub investigator. We gratefully acknowledge the support of DARPA under Nos. FA865018C7880 (ASED), N660011924033 (MCS); ARO under Nos. W911NF-16-1-0342 (MURI), W911NF-16-1-0171 (DURIP); NSF under Nos. OAC-1835598 (CINES), OAC-1934578 (HDR); Stanford Data Science Initiative, Wu Tsai Neurosciences Institute, Chan Zuckerberg Biohub, JD.com, Amazon, Boeing, Docomo, Huawei, Hitachi, Observe, Siemens, UST Global.

The U.S. Government is authorized to reproduce and distribute reprints for Governmental purposes notwithstanding any copyright notation thereon. Any opinions, findings, and conclusions or recommendations expressed in this material are those of the authors and do not necessarily reflect the views, policies, or endorsements, either expressed or implied, of DARPA, NIH, ARO, or the U.S. Government.

The Pande Group acknowledges the generous support of Dr. Anders G. Frøseth and Mr. Christian Sundt for our work on machine learning. The Pande Group is broadly supported by grants from the NIH (R01 GM062868 and U19 AI109662) as well as gift funds and contributions from Folding@home donors.

V.S.P is a consultant & SAB member of Schrodinger, LLC and Globavir, sits on the Board of Directors of Apeel Sciences, Asimov, BioAge Labs, Ciitizen, Devoted Health, Freenome, Insitro, Omada Health, PatientPing, and is a General Partner at Andreessen Horowitz.

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

# A  DETAILS OF GNN ARCHITECTURES

Here we describe GNN architectures used in our molecular property and protein function prediction experiments. For both domains we use the GIN architecture (Xu et al., 2019) with some minor modifications to include edge features, as well as center node information in the protein ego-networks.

As our primary goal is to systematically compare our pre-training strategy to the strong baseline strategies, we fix all of these hyper-parameters in our experiments and focus on relative improvement directly caused by the difference in pre-training strategies.

**Molecular property prediction.** In molecular property prediction, the raw node features and edge features are both 2-dimensional categorical vectors (see Appendix C for details), denoted as $(i_{v,1}, i_{v,2})$ and $(j_{e,1}, j_{e,2})$ for node $v$ and edge $e$, respectively. Note that we also introduce unique categories to indicate masked node/edges as well as self-loop edges. As input features to GNNs, we first embed the categorical vectors by

$$h_v^{(0)} = \text{EmbNode}_1(i_{v,1}) + \text{EmbNode}_2(i_{v,2})$$
$$h_e^{(k)} = \text{EmbEdge}_1^{(k)}(j_{e,1}) + \text{EmbEdge}_2^{(k)}(j_{e,2}) \quad \text{for} \ k = 0, 1, \dots, K - 1,$$

where $\text{EmbNode}_1(\cdot)$, $\text{EmbNode}_2(\cdot)$, $\text{EmbEdge}_1^{(k)}(\cdot)$, and $\text{EmbNode}_1^{(k)}(\cdot)$ represent embedding operations that map integer indices to $d$-dimensional real vectors, and $k$ represents the index of GNN layers. At the $k$-th layer, GNNs update node representations by

$$h_v^{(k)} = \text{ReLU}\left(\text{MLP}^{(k)}\left(\sum_{u \in \mathcal{N}(v) \cup \{v\}} h_u^{(k-1)} + \sum_{e=(v,u):u \in \mathcal{N}(v) \cup \{v\}} h_e^{(k-1)}\right)\right), \qquad \text{(A.1)}$$

where $\mathcal{N}(v)$ is a set of nodes adjacent to $v$, and $e = (v, v)$ represents the self-loop edge. Note that for the final layer, i.e., $k = K$, we removed the ReLU from Eq. (A.1) so that $h_v^{(k)}$ can take negative values. This is crucial for pre-training methods based on the dot product, *e.g.*, Context Prediction and Edge Prediction, as otherwise, the dot product between two vectors would be always positive.

The graph-level representation $h_G$ is obtained by averaging the node embeddings at the final layer, i.e.,

$$h_G = \text{MEAN}(\{h_v^{(K)} \mid v \in G\}). \qquad \text{(A.2)}$$

The label prediction is made by a linear model on top of $h_G$.

In our experiments, we set the embedding dimension $d$ to 300. For MLPs in Eq. (A.1), we use the ReLU activation with 600 hidden units. We apply batch normalization (Ioffe & Szegedy, 2015) right before the ReLU in Eq. (A.1) and apply dropout (Srivastava et al., 2014) to $h_v^{(k)}$ at all the layers except the input layer.

**Protein function prediction.** The GNN architecture used for protein function prediction is similar to the one used for molecular property prediction except for a few differences. First, the raw input node features are uniform (denoted as $X$ here) and second, the raw input edge features are binary vectors (see Appendix D for the detail), which we denote as $c_e \in \{0, 1\}^{d_0}$. As input features to GNNs, we first embed the raw features by

$$h_v^{(0)} = X$$
$$h_e^{(k)} = W c_e + b \quad \text{for} \ k = 0, 1, \dots, K - 1,$$

where $W \in \mathbb{R}^{d \times d_0}$ and $b \in \mathbb{R}^d$ are learnable parameters, and $h_v^{(0)}, h_e^{(k)} \in \mathbb{R}^d$. At each layer, GNNs update node representations by

$$h_v^{(k)} = \text{ReLU}\left(\text{MLP}^{(k)}\left(\text{CONCAT}\left(\sum_{u \in \mathcal{N}(v) \cup \{v\}} h_u^{(k-1)}, \sum_{e=(v,u):u \in \mathcal{N}(v) \cup \{v\}} h_e^{(k-1)}\right)\right)\right),$$
$$\text{(A.3)}$$

| Dataset | | MUTAG | PTC |
|---|---|---|---|
| # Molecules | | 188 | 344 |
| # Binary prediction tasks | | 1 | 1 |
| Prvious results | | Cross validation split | |
| WL substree (Douglas, 2011) | | $90.4 \pm 5.7$ | $59.9 \pm 4.3$ |
| Patchysan (Niepert et al., 2016) | | $92.6 \pm 4.2$ | $60.0 \pm 4.8$ |
| GIN (Xu et al., 2019) | | $89.4 \pm 5.6$ | $64.6 \pm 7.0$ |
| Pre-training strategy | | Cross validation split | |
| Graph-level | Node-level | | |
| – | – | $89.3 \pm 7.4$ | $62.4 \pm 6.3$ |
| – | Infomax | $89.8 \pm 5.6$ | $65.9 \pm 3.9$ |
| – | EdgePred | $91.9 \pm 7.0$ | $66.5 \pm 5.7$ |
| – | Masking | $91.4 \pm 5.0$ | $64.4 \pm 7.3$ |
| – | ContextPred | $92.4 \pm 7.1$ | $68.3 \pm 7.8$ |
| Supervised | – | $90.9 \pm 5.8$ | $64.7 \pm 7.9$ |
| Supervised | Infomax | $90.9 \pm 5.4$ | $63.0 \pm 9.3$ |
| Supervised | EdgePred | $91.9 \pm 4.2$ | $63.5 \pm 8.2$ |
| Supervised | Masking | $90.3 \pm 3.3$ | $60.9 \pm 9.1$ |
| Supervised | ContextPred | $92.5 \pm 5.0$ | $66.5 \pm 5.2$ |

Table 3: **10-fold cross validation accuracy (%) on classic graph classification benchmarks using different pre-training strategies with GIN.** All the previous results are excerpted from Xu et al. (2019).

where $\text{CONCAT}(\cdot, \cdot)$ takes two vectors as input and concatenates them. Since the downstream task is ego-network classification, we use the embedding of the center node $v_{\text{center}}$ together with the embedding of the entire ego-network. More specifically, we obtain graph-level representation $h_G$ by

$$h_G = \text{CONCAT}\left(\text{MEAN}(\{h_v^{(K)} \mid v \in G\}), h_{v_{\text{center}}}^{(K)}\right). \tag{A.4}$$

**Other GNN architectures.** For GCN, GraphSAGE, and GAT, we adopt the implementation in the Pytorch Geometric library (Fey & Lenssen, 2019), where we set the number of GAT attention heads to be 2. The dimensionality of node embeddings as well as the number of GNN layers are kept the same as GIN. These models do not originally handle edge features. We incorporate edge features into these models similarly to how we do it for the GIN; we add edge embeddings into node embeddings, and perform the GNN message-passing on the obtained node embeddings.

## B  EXPERIMENTS ON CLASSIC GRAPH CLASSIFICATION BENCHMARKS

In Table 3 we report our experiments on the commonly-used classic graph classification benchmarks (Kersting et al., 2016). Among the datasets Xu et al. (2019) used, MUTAG, PTC, and NCI1 are molecule datasets for binary classification. Out of these three, we excluded the NCI1 dataset, because it misses edge information (*i.e.*, bond type) and therefore, we cannot recover the original molecule information, which is necessary to construct our input representations described in Appendix C.

For fair comparison, we used exactly the same evaluation protocol as Xu et al. (2019), *i.e.*, report 10-fold cross-validation accuracy. All the hyper-parameters in our experiments are kept the same in the main experiments except that we additionally tuned dropout rate from $\{0, 0.2, 0.5\}$ and the batch size from $\{8, 64\}$ at the fine-tuning stage.

While the pre-trained GNNs (especially those with Context Prediction) give competent performance, all the accuracies (including all the previous methods) are within a standard deviation with each other, making it hard to reliably compare different methods. As Xu et al. (2019) has pointed out, this is due to the extremely small dataset size; a validation set at each fold only contains around 19 to 35 molecules for MUTAG and PTC, respectively. Given these results, we argue that it is necessary to use larger datasets to make reliable comparison, so we mainly focus on MoleculeNet (Wu et al., 2018) in this work.

## C  DETAILS OF MOLECULAR DATASETS

**Input graph representation.** For simplicity, we use a minimal set of node and bond features that unambiguously describe the two-dimensional structure of molecules. We use RDKit (Landrum et al., 2006) to obtain these features.

- Node features:
    - Atom number: [1, 118]
    - Chirality tag: {unspecified, tetrahedral cw, tetrahedral ccw, other}
- Edge features:
    - Bond type: {single, double, triple, aromatic}
    - Bond direction: {–, endupright, enddownright}

**Downstream task datasets.** 8 binary graph classification datasets from Moleculenet (Wu et al., 2018) are used to evaluate model performance.

- **BBBP.** Blood-brain barrier penetration (membrane permeability) (Martins et al., 2012).
- **Tox21.** Toxicity data on 12 biological targets, including nuclear receptors and stress response pathways (Tox21).
- **ToxCast.** Toxicology measurements based on over 600 in vitro high-throughput screenings (Richard et al., 2016).
- **SIDER.** Database of marketed drugs and adverse drug reactions (ADR), grouped into 27 system organ classes (Kuhn et al., 2015).
- **ClinTox.** Qualitative data classifying drugs approved by the FDA and those that have failed clinical trials for toxicity reasons (Novick et al., 2013; AACT).
- **MUV.** Subset of PubChem BioAssay by applying a refined nearest neighbor analysis, designed for validation of virtual screening techniques (Gardiner et al., 2011).
- **HIV.** Experimentally measured abilities to inhibit HIV replication (**?**).
- **BACE.** Qualitative binding results for a set of inhibitors of human $\beta$-secretase 1 (Subramanian et al., 2016).

## D  DETAILS OF PROTEIN DATASETS

**Input graph representation.** The protein subgraphs only have edge features.

- Edge features:
    - Neighbourhood: {True, False}
    - Fusion: {True, False}
    - Co-occurrence: {True, False}
    - Co-expression: {True, False}
    - Experiment: {True, False}
    - Database: {True, False}
    - Text: {True, False}

These edge features indicate whether a particular type of relationship exists between a pair of proteins:

- Neighbourhood: if a pair of genes are consistently observed in each other's genome neighbourhood
- Fusion: if a pair of proteins have their respective orthologs fused into a single protein-coding gene in another organism
- Co-occurrence: if a pair of proteins tend to be observed either as present or absent in the same subset of organisms
- Co-expression: if a pair of proteins share similar expression patterns

- Experiment: if a pair of proteins are experimentally observed to physically interact with each other

- Database: if a pair of proteins belong to the same pathway, based on assessments by a human curator

- Text mining: if a pair of proteins are mentioned together in PubMed abstracts

**Datasets.** A dataset containing protein subgraphs from 50 species is used (Zitnik et al., 2019). The original PPI networks do not have node attributes, but contain edge attributes that correspond to the degree of confidence for 7 different types of protein-protein relationships. The edge weights range from 0, which indicates no evidence for the specific relationship, to 1000, which indicates the highest confidence. The weighted edges of the PPI networks are thresholded such that the distribution of edge types across the 50 PPI networks are uniform. Then, for every node in the PPI networks, subgraphs centered on each node were generated by: (1) performing a breadth first search to select the subgraph nodes, with a search depth limit of 2 and a maximum number of 10 neighbors randomly expanded per node, (2) including the selected subgraph nodes and all the edges between those nodes to form the resulting subgraph.

The entire dataset contains 394,925 protein subgraphs derived from 50 species. Out of these 50 species, 8 species (arabidopsis, celegans, ecoli, fly, human, mouse, yeast, zebrafish) have proteins with GO protein annotations. The dataset contains 88,000 protein subgraphs from these 8 species, of which 57,448 proteins have at least one positive coarse-grained GO protein annotation and 22,876 proteins have at least one positive fine-grained GO protein annotation. For the self-supervised pre-training dataset, we use all 394,925 protein subgraphs.

We define *fine-grained protein functions* as Gene Ontology (GO) annotations that are leaves in the GO hierarchy, and define *coarse-grained protein functions* as GO annotations that are the immediate parents of leaves (Ashburner et al., 2000; Consortium, 2018). For example, a fine-grained protein function is "Factor XII activation", while a coarse-grained function is "positive regulation of protein". The former is a specific type of the latter, and is much harder to derive experimentally. The GO hierarchy information is obtained using GOATOOLS (Klopfenstein et al., 2018). The supervised pre-training dataset and the downstream evaluation dataset are derived from the 8 labeled species, as described in Appendix E. The 40-th most common *fine-grained* protein label only has 121 positively annotated proteins, while the 40-th most common *coarse-grained* protein label has 9386 positively annotated proteins. This illustrates the extreme label scarcity of our downstream tasks.

For supervised pre-training, we combine the train, validation, and prior sets described previously, with the 5,000 most common coarse-grained protein function annotations as binary labels. For our downstream task, we predict the 40 most common fine-grained protein function annotations, to ensure that each protein function has at least 10 positive labels in our test set.

## E    DETAILS OF DATASET SPLITTING

For molecular prediction tasks, following Ramsundar et al. (2019), we cluster molecules by scaffold (molecular graph substructure) (Bemis & Murcko, 1996), and recombine the clusters by placing the most common scaffolds in the training set, producing validation and test sets that contain structurally different molecules. Prior work has shown that this *scaffold split* provides a more realistic estimate of model performance in prospective evaluation compared to random split (Chen et al., 2012; Sheridan, 2013). The split for train/validation/test sets is 80%:10%:10%.

In the PPI network, *species split* simulates a scenario where we have only high-level *coarse-grained* knowledge on a subset of proteins (prior set) in a species of interest (human in our experiments), and want to predict *fine-grained* biological functions for the rest of the proteins in that species (test set). For species split, we use 50% of the protein subgraphs from human as test set, and 50% as a prior set containing only coarse-grained protein annotations. The protein subgraphs from 7 other labelled species (arabidopsis, celegans, ecoli, fly, mouse, yeast, zebrafish) are used as train and validation sets, which are split 85% : 15%. The effective split ratio for the train/validation/prior/test sets is 69% : 12% : 9.5% : 9.5%.

## F    TIME COMPLEXITY OF PRE-TRAINING

Here we analyze the time complexity for processing graphs in Attribute Masking and Context Prediction. First, the time complexity for Attribute Masking is linear with respect to the number of edges/nodes as it only involves sampling nodes/edges to be masked. Second, the time complexity for Context Prediction is again linear with respect to the number of edges/nodes, because it involves sampling a center node per graph plus extracting $K$-hop neighborhood and context graph. Extracting the neighborhood/context graphs is performed by the breadth-first search, which takes at most linear time with respect to the number of edges in the graph. In summary, the time complexity for both of our pre-training methods are at most linear with respect to the number of edges, which is as efficient as message-passing computation in GNNs, and thus, is as efficient as the ordinary supervised learning using GNNs. Also, there is almost no memory overhead as we transform data (e.g., mask input node/edge features, sample the context graphs) on-the-fly.

## G    FURTHER DETAILS OF THE EXPERIMENTAL SETUP

**Optimization.** All models are trained with Adam optimizer (Kingma & Ba, 2015) with a learning rate of 0.001. We use Pytorch (Paszke et al., 2017) and Pytorch Geometric (Fey & Lenssen, 2019) for all of our implementation. We run all pre-training methods for 100 epochs. For self-supervised pre-training, we use a batch size of 256, while for supervised pre-training, we use a batch size of 32 with dropout rate of 20%.

**Fine-tuning.** After pre-training, we follow the procedure in Section 3.3 to fine-tune the models on the training sets of the downstream datasets. We use a batch size of 32 and dropout rate of 50%. Datasets with multiple prediction tasks are fit jointly. On the molecular property prediction datasets, we train models for 100 epochs, while on the protein function prediction dataset (with the 40 binary prediction tasks), we train models for 50 epochs.

**Evaluation.** We evaluate test performance on downstream tasks using ROC-AUC (Bradley, 1997) with the validation early stopping protocol, *i.e.*, test ROC-AUC at the best validation epoch is reported. For datasets with multiple prediction tasks, we take the average ROC-AUC across all their tasks. The downstream experiments are run with 10 random seeds, and we report mean ROC-AUC and standard deviation.

**Computation time for pre-training.** The computation time for the two stages of our pre-training is reported below. **Chemistry:** Self-supervised pre-training takes about 24 hours, while supervised pre-training takes about 11 hours. **Biology:** Self-supervised pre-training takes about 3.8 hours, while supervised pre-training takes about 2.5 hours.

## H    COMPARISON OF PRE-TRAINING WITH DIFFERENT GNN ARCHITECTURES

Table 4 shows the detailed comparison of different GNN architectures on the chemistry datasets. We see that the most expressive GIN architectures benefit most from pre-training compared to the other less expressive models.

## I    ADDITIONAL TRAINING AND VALIDATION CURVES

**Training and validation curves.** In Figure 5, we plot training and validation curves for all the datasets used in the molecular property prediction experiments.

**Additional scatter plot comparisons of ROC-AUCs.** In Figure 6, we compare our Context Prediction + graph-level supervised pre-training with a non-pre-trained model and a graph-level supervised pre-trained model. We see from the left plot that the combined strategy again completely avoids negative transfer across all the 40 downstream tasks. Furthermore, we see from the right plot that additionally adding our node-level Context Prediction pre-training almost always improves ROC-AUC scores of supervised pre-trained models *across the 40 downstream tasks*.

| Dataset | | BBBP | Tox21 | ToxCast | SIDER | ClinTox | MUV | HIV | BACE | Average |
|---|---|---|---|---|---|---|---|---|---|---|
| # Molecules | | 2039 | 7831 | 8575 | 1427 | 1478 | 93087 | 41127 | 1513 | / |
| # Binary prediction tasks | | 1 | 12 | 617 | 27 | 2 | 17 | 1 | 1 | / |
| Configuration | | Out-of-distribution prediction (scaffold split) | | | | | | | | |
| Architecture | Pre-train? | | | | | | | | | |
| GIN | No | 65.8 ±4.5 | 74.0 ±0.8 | 63.4 ±0.6 | 57.3 ±1.6 | 58.0 ±4.4 | 71.8 ±2.5 | 75.3 ±1.9 | 70.1 ±5.4 | 67.0 |
| GIN | Yes | 68.7 ±1.3 | **78.1 ±0.6** | **65.7 ±0.6** | **62.7 ±0.8** | **72.6 ±1.5** | **81.3 ±2.1** | **79.9 ±0.7** | **84.5 ±0.7** | **74.2** |
| GCN | No | 64.9±3.0 | 74.9±0.8 | 63.3±0.9 | 60.0±1.0 | 65.8±4.5 | 73.2±1.4 | 75.7±1.1 | 73.6±3.0 | 68.9 |
| GCN | Yes | **70.6±1.6** | 75.8±0.3 | **65.3±0.1** | 62.4±0.5 | 63.6±1.7 | 79.4±1.8 | 78.2±0.6 | 82.3±3.4 | 72.2 |
| GraphSAGE | No | 69.6±1.9 | 74.7±0.7 | 63.3±0.5 | 60.4±1.0 | 59.2±4.4 | 72.7±1.4 | 74.4±0.7 | 72.5±1.9 | 68.3 |
| GraphSAGE | Yes | 63.9±2.1 | 76.8±0.3 | 64.9±0.2 | 60.7±0.5 | 60.7±2.0 | 78.4±2.0 | 76.2±1.1 | 80.7±0.9 | 70.3 |
| GAT | No | 66.2±2.6 | 75.4±0.5 | 64.6±0.6 | 60.9±1.4 | 58.5±3.6 | 66.6±2.2 | 72.9±1.8 | 69.7±6.4 | 66.8 |
| GAT | Yes | 59.4±0.5 | 68.1±0.5 | 59.3±0.7 | 56.0±0.5 | 47.6±1.3 | 65.4±0.8 | 62.5±1.6 | 64.3±1.1 | 60.3 |

Table 4: **Test ROC-AUC (%) performance on molecular prediction benchmarks with different GNN architectures.** The rightmost column averages the mean of test performance across the 8 datasets. For pre-training, we applied Context Prediction + graph-level supervised pre-training.

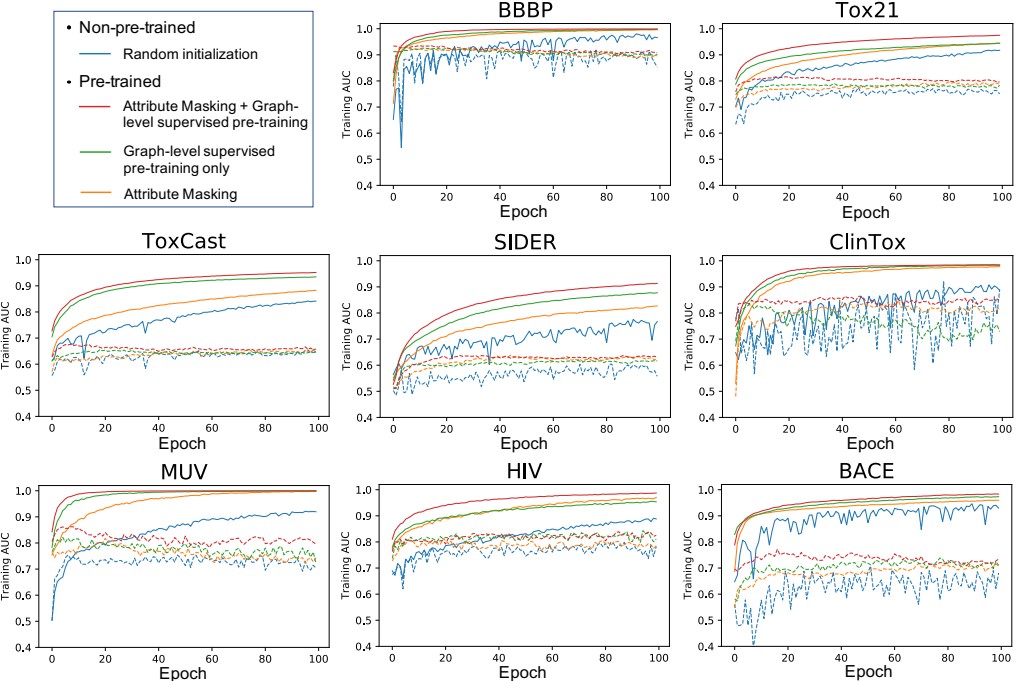

Figure 5: **Training and validation curves of different pre-training strategies.** The solid and dashed lines indicate the training and validation curves, respectively.

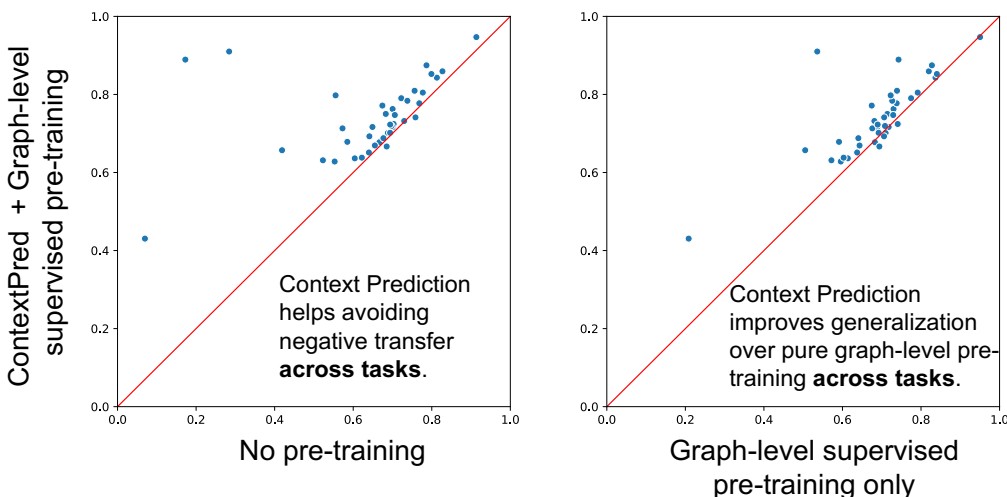

Figure 6: **Scatter plot comparisons** of ROC-AUC scores of our Context Prediction + graph-level supervised pre-training strategy versus the two baseline strategies (non-pre-trained and graph-level supervised pre-trained) on the 40 individual downstream tasks of predicting different fine-grained protein function labels.

