# OpenReview forum: "Strategies for Pre-training Graph Neural Networks"
_ICLR.cc/2020/Conference — Accept (Spotlight)_

### Official Review · AnonReviewer3 · 2019-10-21
**Official Blind Review #3**

**Rating:** 6

**Review:**

The paper proposes pre-training strategies (PT) for graph neural networks (GNN) from both node and graph levels. Two new large-scale pre-training datasets are created and extensive experiments are conducted to demonstrate the benefits of PT upon different GNN architectures. I am relative positive for this work. Detail review of different aspects and questions are as follows.

Novelty: As far as I know, this work is among the earliest works to think about GNN pre-training. The most similar paper at the same period is [Z Hu, arXiv:1905.13728]. I read both papers and found they have similar idea about PT although they have different designs. This paper leverages graph structure (e.g., context neighbors) and supervised labels/attributes (e.g., node attributes, graph labels) for PT. These strategies are not surprising for me and the novelty is incremental.

Experiment: The experiments are overall good. The authors created two new large scale pre-training graph datasets. Experimental results of different GNN architectures w/o different PT for different tasks are provided. Comparing to non-pretraining GNN, the improvements are significant for most cases.

Writing: The writing is good and easy to follow.

Questions: I would like to see more discussion about difference between this work and [Z Hu, arXiv:1905.13728]. Comparing to the other work, what are strengths of this work? In addition, have the authors compared the performances of their work and [Z Hu, arXiv:1905.13728] using the same data?

**Experience Assessment:**

I have published one or two papers in this area.

**Review Assessment: Checking Correctness Of Derivations And Theory:**

N/A

**Review Assessment: Checking Correctness Of Experiments:**

I carefully checked the experiments.

**Review Assessment: Thoroughness In Paper Reading:**

I read the paper thoroughly.

---

> ### Author Response · Authors · 2019-11-08
> **Re: Official Blind Review #3**
>
> We thank the reviewer for acknowledging the novelty of our work and for noting that our experiments are thorough.
>
> Thank you for pointing out a related preprint by Z. Hu et al. [arXiv:1905.13728]. We note the work by Z. Hu et al. was developed independently and concurrently to our work here, and we were not aware of it at the time of writing our paper. We shall cite the preprint and include a discussion in our paper.
>
> Briefly, the key difference between our work and that of Hu et al. is that Hu et al. consider a more restrictive setting where graphs are completely unlabeled (i.e., graphs have no node features). Hu et al. then focus on extracting generic graph properties of unlabeled graphs by pre-training on randomly-generated graphs. While the approach is interesting, the limitation of such an approach is that it improves performance only marginally over ordinary supervised classification of the original attributed graphs. This is because it is hard for random unlabeled graphs to capture domain-specific knowledge that is useful for a specific application. Moreover, in practice, graphs tend to have labels together with rich node and edge attributes, but Hu et al.’s approach cannot naturally leverage such attribute information, which then results in limited gains.
>
> In principle, we could compare our approach against Hu et al., however, right now, this would be extremely challenging because of the following reasons. (1) We cannot find a public implementation of Hu et al.’s approach for reliable comparison. (2) Reimplementing their method requires knowledge of many specific implementational details and design choices (feature extraction, graph generation, etc.), which are not discussed in their preprint. (3) Finally, their pre-trained GNN operates on unlabeled graphs, and so it cannot be directly applied to our datasets of labeled graphs.
>
> Lastly, in contrast to Hu et al., our work focuses on important real-world domains, where one wants to pre-train GNNs by utilizing the abundant graph, node, and edge attributes. Importantly, our approach is able to learn a domain-specific data distribution that is useful for downstream prediction. We demonstrate on two application domains that such practical settings (i.e., labeled graphs with naturally-given node and edge attributes) are very important to consider and that our pre-training can substantially improve model performance.

---

### Official Review · AnonReviewer1 · 2019-10-23
**Official Blind Review #1**

**Rating:** 6

**Review:**

This paper proposes new pre-training strategies for GNN with both a node-level and a graph-level pretraining. For the node-level pretraining, the goal is to map nodes with similar surrounding structures to nearby context (similarly to word2vec). The main problem is that directly predicting the context is intractable because of combinatorial explosion. The main idea is then to use an additional GNN to encode the context and to learn simultaneously the main GNN and the context GNN via negative sampling. Another method used is attribute masking where some masked node and edge attributes need to be predicted by the GNN. For graph-level pretraining, some general graph properties need to be predicted by the graph.
Experiments are conducted on datasets in the chemistry domain and the biology domain showing the benefit of the pre-training.

The paper addresses an important and timely problem. It is a pity that the code is not provided. In particular, the node-level pretraining described in section 3.1.1. seems rather complicated to implement as a context graph needs to be computed for each node in the graph. In particular I do not think the satement 'all the pre-training methods are at most linear with respect to the number of edges' made in appendix F is correct.

**Experience Assessment:**

I have read many papers in this area.

**Review Assessment: Checking Correctness Of Derivations And Theory:**

I assessed the sensibility of the derivations and theory.

**Review Assessment: Checking Correctness Of Experiments:**

I did not assess the experiments.

**Review Assessment: Thoroughness In Paper Reading:**

I read the paper at least twice and used my best judgement in assessing the paper.

---

> ### Author Response · Authors · 2019-11-08
> **Re: Official Blind Review #1**
>
> We thank the reviewer for acknowledging the technical aspects of the paper and for noting that our​ ​results​ ​are​ ​solid​ ​and​ ​our​ ​analysis​ ​is​ ​thorough.
>
> RE: Source code
> The reviewer makes an important point about the availability of the source code. To address this point, in the link privately shared with the reviewers, we have provided all of our code, datasets together with their train/test splits, as well as our pre-trained models, to help with the reproducibility of our results. We note that we will share PyTorch implementations of all pre-training methods and datasets with the community upon publication. Please feel free to ask any further questions regarding our code and implementation.
>
> RE: Linear time complexity in Appendix F
> We acknowledge that the time complexity of our pre-training methods was not well explained in Appendix F. In Figure 2 (a) we show that we only sample one node per graph. We then use breadth-first search to extract a K-hop neighborhood of the node, which takes at most linear time with respect to the number of edges in the graph. As a result, pre-training via context prediction has linear time complexity. We will edit Appendix F to include more detailed information and cover this important point.
>
> Please let us know if you have any further questions or comments!

---

### Official Review · AnonReviewer2 · 2019-10-28
**Official Blind Review #2**

**Rating:** 6

**Review:**

The authors introduce strategies for pre-training graph neural networks. Pre-training is done at the node level as well as at the graph level. They evaluate their approaches on two domains, biology and chemistry on a number of downstream tasks. They find that not all pre-training strategies work well and can in fact lead to negative transfer. However, they find that pre-training in general helps over non pre-training.

Overall, this paper was well written with useful illustrations and clear motivations. The authors evaluate their models over a number of datasets. Experimental construction and analysis also seems sound.

I would have liked to see a bit more analysis as to why some pre-training strategies work over others. However, the authors mention that this is in their planned future work.

Also, in figure 4, the authors mention that their pre-trained models tend to converge faster. However, this does not take into account the time already spent on pre-training. Perhaps the authors can include some results as to the total time taken as well as amortized total time over a number of different downstream tasks.


**Experience Assessment:**

I have published one or two papers in this area.

**Review Assessment: Checking Correctness Of Derivations And Theory:**

I assessed the sensibility of the derivations and theory.

**Review Assessment: Checking Correctness Of Experiments:**

I assessed the sensibility of the experiments.

**Review Assessment: Thoroughness In Paper Reading:**

I read the paper at least twice and used my best judgement in assessing the paper.

---

> ### Author Response · Authors · 2019-11-08
> **Re: Official Blind Review #2**
>
> We thank the reviewer for insightful feedback and for noting that our​ experiments ​are​ ​solid​ and our setup and analyses are sound. The reviewer asks great questions, and we provide the answers below.
>
> RE: Total running time
> The reviewer raises an interesting point about total training time, which includes the time to pre-train a GNN and the time to fine-tune it on a downstream task. To address this point, below, we give the results of the total training time as well as the amortized total time over different downstream tasks. We will include detailed results and a discussion in the final version of the paper.
>
> We note that although pre-training does take some time, it is a one-time-effort only. That is, we pre-train a GNN model only once and then reuse it many times by fine-tuning the model on any number of downstream prediction tasks. Overall, we find that GNNs, once pre-trained, tend to converge much faster on downstream tasks. Most importantly, we find (details below) that validation set performance converges 5-12 times more quickly when GNNs are pre-trained. We emphasize that this cannot be achieved by mere training of (non-pre-trained) GNNs longer. The following summarizes training time for chemistry and biology datasets.
>
> 1) Chemistry dataset (single GPU implementation)
> **Pre-training**
> — Self-supervised pre-training: 24 hours
> — Supervised pre-training: 11 hours
>
> **Fine-tuning on MUV dataset** [Time to achieve the best validation set AUC]
> — From random initialization (i.e., no pre-training): 1 hour; 74.9% AUC
> — From a pre-trained GNN: 5 minutes; 85.3% AUC
>
> 2) Biology dataset
> **Pre-training**
> — Self-supervised pre-training:  3.8 hours
> — Supervised pre-training: 2.5 hours
>
> **Fine-tuning** [Time to achieve the best validation set AUC]
> — From random initialization (i.e., no pre-training): 50 minutes; 84.8% AUC
> — From a pre-trained GNN: 10 minutes; 88.8% AUC
>
> On chemistry dataset, we see that fine-tuning a pre-trained GNN on the MUV required only 5 min. This is in sharp contrast with training a GNN from scratch, which required 12x more time, yet it gave a worse performance. We can reach similar conclusions on the biology dataset. We thus recommend using pre-trained models whenever possible as they can give better performance and can be reused for any number of downstream tasks.
>
> We shall add these results and explanations to the final version of the paper.
>
> RE: Analysis of different pre-training strategies
> Thank you for bringing up this valuable point. We agree that it is important to understand why some pre-training strategies work better over others.  Our key insight backed up with extensive empirical evidence is that a combination of graph-level and node-level methods (Figure 1) is important because it allows the model to capture both local and global semantics of graphs. Further, we find that our structure-based node-level methods (Context Prediction and Attribute Masking) are preferred over position-based node-level methods (Edge Prediction, Deep Graph Infomax). As future work, we plan to further investigate what graph-level and node-level methods are most useful in different domains, and understand what domain-specific knowledge has been learned by the pre-trained models.

---

### Public Comment · ~Junhyun_Lee1 · 2019-10-04
**Available repository for Reproducibility**

Because the pre-trained models are commonly used for various down-stream tasks, I think there should be available URL for codes and pre-trained weights to test its scalability (transferability).

Secondly, "Out-of-distribution prediction (scaffold split)" is conducted at this work, so it would be better if there are URL providing dataset split.

Third, as I found, there are several settings for download of large scale dataset (ZINC), therefore the providing of large scale dataset you used makes this work more reproducible.


* To sum up, could you please provide the URL for codes, large scale datasets, and down-stream datasets (with scaffold split)?

---

> ### Author Response · Authors · 2019-10-08
> **Re: Available repository for Reproducibility**
>
> We thank the reader for his comments and suggestions. We are in total agreement with the suggestions.
> We are working on a comprehensive project website where we will be releasing clean and easy to use data
> together with the splits, which will greatly help the community to move beyond small graph classification benchmarks.
>
> We are also working on releasing the code as well as the final pre-trained models so that the community
> can benefit from this work and use the models/data for other downstream prediction tasks.

---

> > ### Public Comment · ~Junhyun_Lee1 · 2019-10-14
> > **Re: Re: Available repository for Reproducibility**
> >
> > Thank you for your reply.
> >
> > These contributions will bring many benefits to the communities (ML, Bio, and Chemical).
> > Because the pre-training GNNs is really essential but was a challenging problem.
> >
> > I am looking forward to the available URLs for data and models.
> >
> > Nice work!

---

### Public Comment · ~Simon_Shaolei_Du1 · 2019-11-08
**Related Work**

Dear authors,
This is a very interesting paper! We would like to draw your attention to our recent paper: https://arxiv.org/abs/1905.13192
On PTC, our graph neural tangent kernel achieves 67.9, which is the best result to date (we are aware of).

---

### Decision · Program_Chairs · 2019-12-19

**Decision:**

Accept (Spotlight)

**Comment:**

All three reviewers are consistently positive on this paper. Thus an accept is recommended.